# Heightened Willingness toward Pneumococcal Vaccination in the Elderly Population in Shenzhen, China: A Cross-Sectional Study during the COVID-19 Pandemic

**DOI:** 10.3390/vaccines9030212

**Published:** 2021-03-03

**Authors:** Minyi Zhang, Hongbiao Chen, Fei Wu, Qiushuang Li, Qihui Lin, He Cao, Xiaofeng Zhou, Zihao Gu, Qing Chen

**Affiliations:** 1Department of Epidemiology, School of Public Health, Southern Medical University, Guangzhou 510515, China; zhangminyi1993@outlook.com (M.Z.); wufei1996123@outlook.com (F.W.); li15938303617@163.com (Q.L.); 2Department of Epidemiology and Infectious Disease Control, Longhua Centre for Disease Control and Prevention, Shenzhen 518109, China; lhjbkz@163.com (H.C.); lhjkzx@szlhq.gov.cn (Q.L.); luojingwei001@163.com (H.C.); ct502528528@163.com (X.Z.); 13711762051@163.com (Z.G.); 3Department of Epidemiology and Infectious Disease Control, Longhua Key Discipline of Control and Prevention of Infectious Diseases and Public Health, Shenzhen 518109, China

**Keywords:** 23-valent pneumococcal polysaccharide vaccine, older adults, positive attitude, influencing factors, COVID-19

## Abstract

Background: Elderly population is considered at high risk for pneumococcal diseases. The pneumococcal vaccine coverage presents extremely low among elderly people in China. However, the serious event of COVID-19 drives interest in the pneumococcal vaccine, prompting us investigating the willingness to accept the 23-valent pneumococcal polysaccharide vaccine (PPSV23) and influencing factors among people aged over 60 years during the COVID-19 pandemic. Methods: A cross-sectional study was employed using a self-administered questionnaire in Shenzhen City of China, elaborating the willingness toward PPSV23 in the elderly persons. Binomial logistic analyses were performed to estimate the influencing factors using odds ratios (ORs) and 95% confidence interval (CI). Results: Among 15,066 respondents, 91.5% presented a positive attitude toward PPSV23. Logistic analyses suggested the influencing factors included knowledge about pneumonia (adjusted OR [aOR] 1.391, 95%CI 1.214–1.593), perception of the seriousness of pneumonia (aOR 1.437, 95%CI 1.230–1.680) and preventing way for pneumonia (aOR 1.639, 95%CI 1.440–1.865), worried about getting pneumonia (aOR 2.751, 95%CI 2.444–3.096), understanding vaccine policy (aOR 1.774, 95%CI 1.514–2.079), and influenza vaccine (aOR 3.516 and 95%CI 2.261–5.468) and PPSV23 histories (aOR 3.199, 95%CI 1.492–6.860). Conclusions: The interest surge in pneumococcal vaccine coincided with the COVID-19 outbreak, foreshadowing higher demand for pneumococcal vaccine in the near future.

## 1. Introduction

As the most common and life-threatening infection, pneumococcal pneumonia results in the primary cause of morbidity and mortality in the elderly population globally [1,2]. The disease burden of hospitalizations and deaths related to pneumococcal pneumonia increased rapidly in countries with an aging population [3,4]. Among the pathogens, *Streptococcus pneumoniae* ranks as the most common causative agent of community-acquired pneumonia (CAP) in adults [5]. In China, a study most recently reported the incidence rates of CAP increased with age in adults, ranging from 7.8 to 14.9 per 1000 person-years among people aged over 60 years [6]. Other studies had reported different proportions of pneumococcal infections in CAP cases in different regions of China, varying from 28.0% to 71.5% [7,8,9].

The China Antimicrobial Resistant Surveillance System revealed the highly antimicrobial resistance rates regarding *Streptococcus pneumoniae* in elderly people aged ≥65 years, with a level of 46.4%, 53.6%, and 94.2% for cefuroxime, penicillin, and erythromycin, respectively [10]. Accordingly, vaccination plays an influential role in preventing pneumococcal pneumonia. Based on global efforts, pneumococcal vaccinations have developed rapidly in the past few decades. The elderly individuals at high risk of being infected should be prioritized for pneumococcal vaccination. At present, two vaccines have been widely used for pneumonia protection, including the 13-valent pneumococcal conjugate vaccine (PCV13) and the 23-valent pneumococcal polysaccharide vaccine (PPSV23), and the latter was approved for application in elderly persons according to Chinese guidelines for vaccination [11]. However, the vaccination coverage with PPSV23 remains exceptionally low in most areas of mainland China [12] when compared to Hong Kong [13] and other developed countries, such as the United States and England, with the pneumococcal vaccination coverage as 61.3% and 69.8%, respectively for older adults [1,14]. Furthermore, the pneumococcal vaccine coverage rates differ in different age groups. In China, a previous report on vaccination status of PPSV23 by age highlighted the PPSV23 coverage rate varied with years, indicating higher rates in children under 14 years old ranged from 12.14% to 53.94%; and however, it was less than 1.0% among people aged over 15 years in 2014 [15].

A prior study indicated that vaccination decisions, public awareness of vaccination, economic barriers, vaccine or disease-related knowledge, perceived vulnerability, and seriousness of diseases are closely equated to the vaccination practices [16]. Likewise, merely 21.8% of urban elderly people in China manifested positive attitudes toward PPSV23 vaccination [17]. What’s more, only 38.7% of individuals with rheumatic diseases in China displayed a willingness to pneumococcal vaccine uptake under the recommendation from doctors [18]. Hence, there is an urgent request to raise public awareness of vaccination and immunization [19].

In general, public interest in vaccinations and health-seeking behavior during outbreaks may increase concerning diseases that expressed similar symptoms to the outbreak [20,21,22,23]. An infodemiology study has confirmed a peak in global interest in pneumococcal vaccine matched up with the COVID-19 pandemic between February and March 2020 [24]. Since October 2016, the government of Shenzhen City, China, has implemented a program providing free access to pneumococcal vaccination (PPSV23) for residents aged 60 years or older with the Shenzhen census registry or Shenzhen social health insurance. We herein aimed to survey the willingness toward pneumococcal vaccine usage and its influencing factors among the elderly population during the COVID-19 pandemic.

## 2. Materials and Methods

### 2.1. Study Sites and Participants

The present study belonged to a cross-sectional investigation utilizing a self-administered questionnaire, performed in Shenzhen City, with over 20 million residents in southern China. The data were collected through an approved questionnaire from Longhua Center for Disease Control and Prevention (CDC), Shenzhen. Regarding the targeted population, there are a total of 125,598 people aged over 60 years on this study site. Similar to multiple online cross-sectional surveys conducted during the outbreak of COVID-19 [15,25], we could not carry out face-to-face interviews for the participants because of the social restrictions. To collect as many samples as possible, we sent the electric questionnaires to the primary and secondary schools covering the whole of Longhua, an urban district in Shenzhen city. According to our guideline, the parents of the students were responsible for obtaining information on older adults aged over 60 years in their families; and it was unnecessary to fill in the questionnaire repeatedly if there are multiple students in the same family. We believed that older adults were more willing to cooperate with this survey when interviewed by their family members, and thereby the provided contents would be more reliable. Only those older adults who gave consent to be a part of the present study would be recruited as participants. Finally, 15,066 older adults with available information were enrolled in the current survey in October 2020. The respondents were under quality control, and there was no missing data from the responses. We conducted this survey on Wen Juan Xing (Changsha Ranxing Information Technology Co., Ltd., Changsha, Hunan, China), the largest online survey platform in China. Only the completed questionnaires would be submitted successfully. The protocol for this research was approved by the ethics committee of Longhua CDC, Shenzhen.

### 2.2. Questionnaire

We developed a self-administered questionnaire for the current study, and the validity of the questionnaire was assessed by experts in epidemiology and infectious diseases. Cronbach’s alpha was generated using the coefficient test, indicating a score of almost 0.70 (0.69 in the present study). The questionnaire consisted of four parts, with a total of 33 questions covering baseline information, health-related behaviors, and vaccine status. Part I contained the socio-demographic characteristics of the subjects, such as age, gender, marital status, education level, living status, and household income. Part II comprised questions about self-reported health information, such as frequency of health examination, smoking, drinking, excise, chronic disease histories of hypertension, diabetes, and chronic respiratory diseases such as bronchial asthma, bronchitis, chronic obstructive pulmonary disease and pneumoconiosis. Part III focused on influencing factors regarding the knowledge and perception toward pneumonia and the histories of the pneumococcal vaccine and seasonal influenza vaccine. Part IV collected information on the willingness to uptake the pneumococcal vaccine. All questions were closed-ended and treated as categorical variables.

### 2.3. Statistical Analysis

The participants were separated into two groups according to their willingness to pneumococcal vaccination. Descriptive statistics were conducted for the general characteristics and health information of the participants, and their knowledge and perception of pneumonia, as well as the willingness to accept the pneumococcal vaccine. The information of monthly household income was presented in Chinese yuan (CNY) and US dollars (USD) with an average exchange rate of CNY 6.9 per dollar in 2020. The categorical variables are presented as frequencies with percentages and were compared with chi-squared tests or Fisher’s exact tests. Binomial logistic regression models were carried out to explore the associations of knowledge and perception factors with the willingness of pneumococcal vaccination by calculating odds ratios (ORs) and 95% confidence interval (CI). Model I quantified the correlation via univariate analysis without adjustment. Further, the least absolute shrinkage and selection operator (LASSO) method was used for variables selection to fit an optional model via cross-validation. The selected variables were considered confounders and were brought into multivariate analysis (Model II), including age, gender, marital status, education level, monthly household income, frequency of health examination, smoking, excise, and histories of chronic respiratory diseases, hypertension, and diabetes.

All analyses were performed in R statistical software v.4.0.3 (R Foundation for Statistical Computing, Vienna, Austria). We considered statistical tests with *p*-values < 0.05 statistically significant. All *p*-values were two-sided.

## 3. Results

### 3.1. General Characteristics and Health Information of the Subjects

A total of 15,066 subjects aged 60 years and older were involved in the present survey in October 2020. The descriptive characteristics and their willingness to pneumococcal vaccination are shown in Table 1. Of these, 55.0% were female, and most participants aged 60 to 69 years (80.1%), whereas the group aged over 80 years accounted for only 2.7% of subjects. Thirteen thousand seven hundred and seventy-eight (91.5%) displayed a positive attitude toward the pneumococcal vaccination. Those who expressed willingness to uptake the pneumococcal vaccine were more likely to be lower age, married, higher education level, living with children, having a monthly household income of USD 1449–7246, less frequency of health examination, non-smokers, and with a history of chronic respiratory diseases. However, gender was not statistically significant between these two groups.

The subjects were asked to self-report their health information at baseline. Of these, most of subjects (72.1%) reported the frequency of health examination was less than once per year and never smoking (80.9%), whereas relatively few people (9.1%) revealed that they had a history of chronic respiratory diseases. The willingness to receive pneumococcal vaccine was statistically significant between the groups with a different frequency of health examination (*p* < 0.001), status (*p* < 0.001), and chronic respiratory disease history (*p* < 0.001). During the survey, 79.6% revealed they were never drinking (*p* = 0.414), and most of them (78.9%) exercise for more than half an hour per day (*p* = 0.174). Moreover, 34.6% and 11.4% of subjects reported they had hypertension (*p* = 0.330) and diabetes histories (*p* = 0.700), respectively (Table 1).

### 3.2. Knowledge, Perception and Willingness toward Pneumonia and Vaccinations

According to our investigation, numerous participants (81.3%) knew about pneumonia (*p* < 0.001), and a larger proportion of subjects (87.5%) agreed that pneumonia is a severe disease (*p* < 0.001). Moreover, 62.8% reported that they were worried about getting pneumonia (*p* < 0.001). Among the older adults, 38.3% believed that *Streptococcus pneumoniae* could result in pneumonia, sepsis, and meningitis (*p* = 0.748). Meanwhile, 38.7% of them agreed the older adults are susceptible to pneumonia caused by *Streptococcus pneumoniae* infection (*p* = 0.351). Of the subjects, 37.8% believed that pneumococcal vaccination is useful for preventing streptococcus pneumonia in older people (*p* < 0.001), and 22.8% had understood the local pneumococcal vaccination policy (*p* < 0.001). In terms of vaccination histories, few participants (4.7%) had experience with seasonal influenza vaccine (*p* < 0.001) and 1.4% had received pneumococcal vaccine in the past five years (*p* = 0.007) (Table 2).

### 3.3. Factors Associated with the Willingness to Accept Pneumococcal Vaccine in the Elderly Population

We investigated the influencing factors associated with the willingness to pneumococcal vaccine usage, which relates to knowledge and perception for pneumonia and vaccinations. It noted that similar results were observed in the univariate and multivariate analyses (Table 3). As expected, participants who knew better about pneumonia were 1.39 times more likely to get the pneumococcal vaccine than participants who failed to understand (aOR 1.391, 95%CI 1.214–1.593). The perception that pneumonia is a severe disease (aOR 1.437, 95%CI 1.230–1.680) and pneumococcal vaccination is sufficient to prevent pneumonia (aOR 1.639, 95%CI 1.440–1.865) were related to the positive attitude toward pneumococcal vaccination. In parallel, participants concerned about getting pneumonia and knew the local pneumococcal vaccination policy had adjusted ORs of 2.751 (2.444–3.096) and 1.774 (1.514–2.079). Moreover, a history of seasonal influenza vaccine was considered the most factor with an adjusted OR 3.516 and 95%CI 2.261–5.468; meanwhile, those subjects with an experience of pneumococcal vaccination in the past five years also tended to accept pneumococcal vaccine in our study (aOR 3.199, 95%CI 1.492–6.860). Nevertheless, the agreement that *Streptococcus pneumoniae* leads to pneumonia, sepsis, and meningitis (aOR 1.040, 95%CI 0.920–1.175), and older adults are susceptible to pneumonia (aOR 1.091, 95%CI 0.966–1.233) did not contribute to the willingness of pneumococcal vaccination.

## 4. Discussion

In this cross-sectional survey, we elaborated on the willingness of pneumococcal vaccination among Chinese elderly people during the COVID-19 pandemic period. Further, we investigated the knowledge and perception factors associated with the willingness to accept pneumococcal vaccine. It is interesting to highlight that a major of participants (91.5%) stated a positive attitude toward the pneumococcal vaccination, whereas few Chinese older adults (1.4%) have covered pneumococcal vaccine in the present study. Knowledge about pneumonia, perception of the severity and effective prevention method for pneumonia, worried about getting pneumonia, understanding local pneumococcal vaccine policy, and experience with seasonal influenza and pneumococcal vaccines were significantly associated with a higher likelihood of pneumococcal vaccination among the elderly persons.

Compared to prior published studies of China and other countries, the coverage rate of pneumococcal vaccine in the present study was higher than the result in 2017 in Hangzhou, China, where 0.31% of persons aged 60 years and older had received pneumococcal vaccine [15]. Nevertheless, as already mentioned, it was significantly lower than that in the United States in 2014, presenting 61.3% of adults aged over 65 years had vaccinated PPSV23 [14]. The reasons for the different coverage rates between China and the United States might be the different recommendations for vaccine uptake and had been detailed previously [15]. Pneumococcal vaccination plays an essential role in preventing pneumococcal diseases among groups at increased risk of pneumococcal diseases, including very young children, older adults, and the immunocompromised population [26]. Hence, further relevant surveys in terms of vaccination behaviors should be performed in such high-risk groups.

The barriers to pneumococcal vaccine uptake found in our study were in line with previous studies, suggesting that poor understanding about pneumonia and its vaccine, as well as failure to recognize the seriousness of the pneumococcal disease have primarily hindered the attitudes toward pneumococcal vaccination among elderly persons [1,13]. We also explored the attitudes toward pneumococcal vaccination between different subgroups using univariate analysis (data not shown). It found that 76.6% of older adults who lived with children exhibited a willingness to uptake pneumococcal vaccine, significantly different from those who live alone. Living with others might lead to frequent social contact, making these elderly persons more likely to obtain vaccine or health-related information [27]. Besides, the willingness to pneumococcal vaccination might vary by age, marital status, education level, household income, frequency of health examination, smoking, and a history of chronic respiratory diseases.

Vaccinations represent a tremendous contribution to global health. Since the initial clinical trial of a whole-cell heat-killed pneumococcal vaccine was carried out among several thousand gold miners in South Africa between 1911 and 1912 [13,28], the global efforts on pneumococcal vaccine studies had undergone for a century [29]. In China, two pneumococcal vaccines, PCV13 and PPSV23, have been circulated currently targeting different age-groups. The Chinese Food and Drugs Administration (CFDA) approved PCV13 for use in infants and children in 2016 [30], whereas PPSV23 received regulatory approval and are available to Chinese people aged over 60 according to the updated guideline from the Chinese Preventive Medicine Association (CPMA) in 2012 [11]. As mentioned before, the government of Shenzhen has published a policy providing the free pneumococcal vaccine to local elderly people. However, the coverage of pneumococcal vaccine indicated in the present study is not optimistic and far away from our expectations. Such a policy had been carried out in multiple cities of China, such as Beijing, Shanghai, and Chengdu. In contrast, a retrospective cohort study during the policy implementation period in Shanghai reported a pneumococcal vaccine coverage of 22.8% among individuals aged over 15 years with chronic diseases [12]. Moreover, a cross-sectional study involved 235 patients with rheumatic diseases, suggesting none (0.0%) of participants had experience with pneumococcal vaccination [18]. These variable rates might be due to the different population and sampling methods. The reasons for exceptionally low coverage of pneumococcal vaccine are complicated and have been detailed in numerous prior studies, generally containing poor knowledge and awareness of vaccines, lack of recommendation from physicians, and loss of trust in vaccine program because of negative news [12,13,18,27,31].

What’s more, our investigation on the willingness of pneumococcal vaccination was 4.23 times higher than a previous study conducted in urban districts of Hangzhou City, China, between July and September 2013 [17]. As described previously, it was also much higher than a cross-sectional survey reporting 38.7% of patients with rheumatic diseases displayed positive attitudes toward pneumococcal vaccination though under recommendations from doctors [18]. This finding should be noticed and worthy of in-depth exploration. As described before, health-seeking behaviors probably increase regarding illnesses with symptoms similar to the pandemic [20,21,22,23]. Several prior studies demonstrated much more health-seeking behaviors and patient visits worldwide under the H1N1 pandemic situation than during non-pandemic periods among individuals with influenza-like symptoms [20,21,22,23]. Meanwhile, an infodemiology study recently found that since coronavirus disease 2019 (COVID-19) has been confirmed as a Public Health Emergency of Concern in 2020, the public awareness of health increased primarily regarding symptoms that similar to those of COVID-19, including influenza, pneumococcal pneumonia, and their vaccines [24]. Further, we also observed similar results for the willingness to receive the seasonal influenza vaccine (data not shown), in line with a prior study that revealed the pneumococcal vaccine acceptance strongly predicts influenza vaccine usage [12]. Consequently, our finding points to the heightened willingness of pneumococcal vaccination might be due to the increasing public awareness of health conditions during the COVID-19 pandemic period. Although the low level of pneumococcal vaccine coverage in the present study, public awareness and perception may contribute to higher demand for pneumococcal vaccination during this period. The government or community should thereby carry out more health campaigns to improve immunization rates in the near future. Firstly, the government should cooperate with the communities and the units where the elderly live to strengthen the publicity of the pneumococcal vaccine and its benefits for health. Secondly, knowledge about pneumococcal pneumonia and its preventive measures should be distributed to every family in a booklet because the family members might encourage older people to accept pneumococcal vaccines.

Despite the global event of COVID-19 changes in health-seeking behaviors and public demand for the pneumococcal vaccine, a few participants refused acceptance of the pneumococcal vaccine, prompting us further investigated the possible wherefores. It found that physical discomfort, no need for vaccination, insufficient knowledge about pneumonia, doubts concerning the effectiveness of the pneumococcal vaccine, vaccination contraindications, vaccination unsafe, and high cost were responsible for the unwillingness to take the pneumococcal vaccine. Approximately half of the individuals believed they were always under healthy conditions and not necessary to receive pneumococcal vaccine, whereas few subjects worried about the pneumococcal vaccination contraindications. These coincided with several prior studies in China that explored the barriers with regard to the low vaccination rate [12,17,18]. Effective interventions and strategies are required for strengthening the knowledge and perception about pneumonia and its vaccine among the elderly population. We are considering a follow-up study focused on the vaccine coverage rate among elderly persons in the coming years to estimate whether this willingness translates into a surge in pneumococcal vaccination.

There are some limitations exited in the present study. Firstly, the study population only consisted of older adults from a single center that was not representative of all elderly persons in China. Secondly, the collected information might be subject to recall bias due to the utilization of self-reported questionnaires.

## 5. Conclusions

This large population-based study reflected a high level of the willingness to accept pneumococcal vaccine in elderly persons aged over 60 years in Shenzhen, China during the COVID-19 pandemic period. Despite the extremely low level of vaccine coverage to date, we interestingly discovered that more than 90% of the elderly respondents intend to receive pneumococcal vaccine, compared to approximately 30% before the outbreak of COVID-19. The combined findings suggest the epidemic of infectious diseases can promote public awareness of disease prevention. Great efforts are warranted for responding to the higher demand for pneumococcal vaccine in the near future.

## Figures and Tables

**Table 1 vaccines-09-00212-t001:** General characteristics and health information of subjects according to the willingness to accept pneumococcal vaccine, Shenzhen, October 2020.

	N (%)	Willingness to Accept Pneumococcal Vaccine		*p*-Value
		Yes	No	
Total	15,066	13,778 (91.5%)	1288 (8.5%)	
Age group (y)				<0.001
60–69	12,063 (80.1%)	11,113 (80.7%)	950 (73.8%)	
70–79	2602 (17.3%)	2322 (16.9%)	280 (21.7%)	
≥80	401 (2.7%)	343 (2.5%)	58 (4.5%)	
Gender				0.181
Male	6786 (45.0%)	6183 (44.9%)	603 (46.8%)	
Female	8280 (55.0%)	7595 (55.1%)	685 (53.2%)	
Marital status				<0.001
Married	11,454 (76.0%)	10,549 (76.6%)	905 (70.3%)	
Unmarried	29 (0.2%)	27 (0.2%)	2 (0.2%)	
Divorce	261 (1.7%)	232 (1.7%)	29 (2.3%)	
Widowed	3322 (22.0%)	2970 (21.6%)	352 (27.3%)	
Education level				0.015
Middle school or lower	10,545 (70.0%)	9600 (69.7%)	945 (73.4%)	
High school	3722(24.7%)	3446 (25.0%)	276 (21.4%)	
College or above	799 (5.3%)	732 (5.3%)	67 (5.2%)	
Living status				<0.001
Living with partner	3178 (21.1%)	2871 (20.8%)	307 (23.8%)	
Living with children	11,438 (75.9%)	10,548 (76.6%)	890 (69.1%)	
Living alone	450 (3.0%)	359 (2.6%)	91 (7.1%)	
Household income				0.002
<CNY 10,000 (USD 1449)	4940 (32.8%)	4472 (32.5%)	468 (36.3%)	
CNY 10,000–50,000 (USD 1449–7246)	8103 (53.8%)	(54.2%)	632 (9.1%)	
>CNY 50,000 (USD 7246)	2023 (13.4%)	1835 (13.3%)	188 (14.6%)	
Frequency of health examination				<0.001
<1	10,858 (72.1%)	9995 (72.5%)	863 (67.0%)	
1–2	3931 (26.1%)	3554 (25.8%)	377 (29.3%)	
≥3	277 (1.8%)	229 (1.7%)	48 (3.7%)	
Smoking				<0.001
No	12,195 (80.9%)	11,207 (81.3%)	988 (76.7%)	
Yes	2871 (19.1%)	2571 (18.7%)	300 (23.3%)	
Drinking				0.414
No	11,993 (79.6%)	10,979 (79.7%)	1014 (78.7%)	
Yes	3073 (20.4%)	2799 (20.3%)	274 (21.3%)	
Excise				0.174
No	3181 (21.1%)	2890 (21.0%)	291 (22.6%)	
Yes	11,885 (78.9%)	10,888 (79.0%)	997 (77.4%)	
Chronic respiratory diseases				<0.001
No	13,071 (90.9%)	12,488 (90.6%)	1213 (94.2%)	
Yes	1365 (9.1%)	1290 (9.4%)	75 (5.8%)	
Hypertension				0.330
No	9850 (65.4%)	8992 (65.3%)	858 (66.6%)	
Yes	5216 (34.6%)	4786 (34.7%)	430 (33.4%)	
Diabetes				0.700
No	13,344 (88.6%)	12,199 (88.5%)	1145 (88.9%)	
Yes	1722 (11.4%)	1579 (11.5%)	143 (11.1%)	

**Table 2 vaccines-09-00212-t002:** Knowledge and perception for pneumonia and vaccinations according to the willingness to accept pneumococcal vaccine in the elderly population.

Question		N (%)	Willingness to Accept Pneumococcal Vaccine		*p*-Value
			Yes	No	
Have you known about pneumonia					<0.001
No		2823 (18.7%)	2505 (18.2%)	318 (24.7%)	
Yes		12,243 (81.3%)	11,273 (81.8%)	970 (75.3%)	
Pneumonia is a serious disease in older adults					<0.001
No		1885 (12.5%)	1664 (12.1%)	221 (17.2%)	
Yes		13,181 (87.5%)	12,114 (87.9%)	1067 (82.8%)	
Pneumococcal vaccine is effective for preventing *Streptococcus pneumoniae* in elderly people					<0.001
No		9376 (62.2%)	8457 (61.4%)	919 (71.4%)	
Yes		5690 (37.8%)	5321 (38.6%)	369 (28.6%)	
*Streptococcus pneumoniae* leads to pneumonia, sepsis, and meningitis					0.748
No		9295 (61.7%)	8495 (61.7%)	800 (62.1%)	
Yes		5771 (38.3%)	5283 (38.3%)	488 (37.9%)	
Older adults are susceptible to pneumonia					0.351
No		9234 (61.3%)	8429 (61.2%)	805 (62.5%)	
Yes		5832 (38.7%)	5349 (38.8%)	483 (37.5%)	
Would you worry about getting pneumonia					<0.001
No		5607 (37.2%)	4828 (35.0%)	779 (60.5%)	
Yes		9459 (62.8%)	8950 (65.0%)	509 (39.5%)	
Have you known the local pneumococcal vaccination policy					<0.001
No		11,634 (77.2%)	10,546 (76.5%)	1088 (84.5%)	
Yes		3432 (22.8%)	3232 (23.5%)	200 (15.5%)	
Have you ever received the seasonal influenza vaccine					<0.001
No		14,353 (95.3%)	13,086 (95.0%)	1267 (98.4%)	
Yes		713 (4.7%)	692 (5.0%)	21 (1.6%)	
Have you received the pneumococcal vaccine in the past 5 years					0.007
No		14,859 (98.6%)	13,578 (98.5%)	1281 (99.5%)	
Yes		207 (1.4%)	200 (1.5%)	7 (0.5%)	

**Table 3 vaccines-09-00212-t003:** Factors associated with the willingness to accept pneumococcal vaccine in the elderly population.

Question	OR	Model I ^a^	*p*-Value	OR	Model II ^b^	*p*-Value
		95%CI			95%CI	
Have you known about pneumonia						
No	ref.			ref.		
Yes	1.475	1.291–1.687	<0.001	1.391	1.214–1.593	<0.001
Pneumonia is a serious disease in older adults						
No	ref.			ref.		
Yes	1.508	1.293–1.758	<0.001	1.437	1.230–1.680	<0.001
Pneumococcal vaccine is effective for preventing *Streptococcus pneumoniae* in elderly people						
No	ref.			ref.		
Yes	1.567	1.382–1.777	<0.001	1.639	1.440–1.865	<0.001
*Streptococcus pneumoniae* leads to pneumonia, sepsis, and meningitis						
No	ref.			ref.		
Yes	1.020	0.906–1.147	0.748	1.040	0.920–1.175	0.534
Older adults are susceptible to pneumonia						
No	ref.			ref.		
Yes	1.058	0.940–1.190	0.351	1.091	0.966–1.233	0.161
Would you worry about getting pneumonia						
No	ref.			ref.		
Yes	2.837	2.524–3.189	<0.001	2.751	2.444–3.096	<0.001
Have you known the local pneumococcal vaccination policy						
No	ref.			ref.		
Yes	1.667	1.427–1.948	<0.001	1.774	1.514–2.079	<0.001
Have you ever received the seasonal influenza vaccine						
No	ref.			ref.		
Yes	3.190	2.059–4.944	<0.001	3.516	2.261–5.468	<0.001
Have you received the pneumococcal vaccine in the past 5 years						
No	ref.			ref.		
Yes	2.969	1.266–5.740	0.010	3.199	1.492–6.860	0.003

OR, odds ratio; CI, confidence interval. ^a^ Model I was not adjusted. ^b^ Model II was after adjustment for age, gender, marital status, education level, household income, examination, smoking, excise, chronic respiratory diseases, hypertension, and diabetes.

## Data Availability

Access to the data presented in this study can be acquired by connecting to the corresponding authors via e-mail. The data are not publicly available due to restrictions of privacy.

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
