# Peer review of "Heightened Willingness toward Pneumococcal Vaccination in the Elderly Population in Shenzhen, China: A Cross-Sectional Study during the COVID-19 Pandemic"

_vaccines, 2021, doi:10.3390/vaccines9030212_

Round 1

Reviewer 1 Report

Authors investigated population willingness toward pneumococcal vaccination among the people aged > 60 years during covid-19 pandemic. Such studies provide inputs to have a 360 degree view of any pandemic and its impact on other common causative illnesses. The introduction was quite informative and method employed are appropriate to collect valuable information from a population size of > 10,000. It will be beneficial to do a follow-up study in coming years to understand whether this (willingness) translates in to a spike in pneumococcal vaccination.

Author Response

Point: It will be beneficial to do a follow-up study in coming years to understand whether this (willingness) translates into a spike in pneumococcal vaccination.

Response: Thanks for your comments and suggestions that improve our manuscript and play an essential role in our future effort. We fully agree that it is of great importance to conduct a follow-up study focused on pneumococcal vaccine usage in the coming years. Hence, we are going to carry out a relevant survey among elderly persons in the same study site.

Reviewer 2 Report

You wrote: “a study most recently reported the incidence rates of CAP increased with age in adults, ranging from 7.8 to 14.9 per 1000 person-years among people aged over 60 years [6], while others had reported the proportions of CAP varied from 28.0% to 71.5% in different regions in the past decade [7-9].”

[please explain what the last phrase means]

You wrote: “The China Antimicrobial Resistant Surveillance System revealed the highly antimicrobial resistance rates regarding Streptococcus pneumoniae in elderly people aged ≥ 65 years”

[this is important, please provide details of antibiotics and resistance rates]

You wrote: “PPSV23 remains exceptionally low in most areas of mainland China [12] when compared to Hong Kong [13]and other developed countries, such as the United States and England with the pneumococcal vaccination coverage as 61.3% and 69.8%, respectively for older adults [1, 14]”

[pleased report by age deciles as the rate varies by deciles]

You wrot4e: “data was collected through an approved questionnaire from Long”

[Latin: datum is singular, data are plural. Please correct]

You write: “We believed that older adults were more willing to cooperate with this survey when interviewed by their family members, and thereby the provided contents would be more reliable. Only those older adults who gave consent to be a part of the present study would be recruited as participants. Finally, 88 15,066 older adults with available information were enrolled in the current survey in October 2020.”

[This is an excellent idea. How many students did you contact, and how many results came back per student: how many grandparents, uncles , aunts… ? How many refusals, and how many responses had missing data?]

You write: “We developed a self-administered questionnaire which has achieved agreements from epidemiological experts for the current study. Cronbach’s alpha was generated using the coefficient test, indicating a score of almost 0.70 (0.69 in the present study).”

[on which questions was there disagreement?]

What advice do you propose to the government authorities to increase the effectiveness of vaccination campaign programmes and uptake?]

Author Response

Thanks for your comments and suggestions that are important to improve our manuscript and play an essential role in our future effort. Please see our point-by-point response below.

Point 1: The authors should explain what the last phrase means "a study most recently reported the incidence rates of CAP increased with age in adults, ranging from 7.8 to 14.9 per 1000 person-years among people aged over 60 years, while others had reported the proportions of CAP varied from 28.0% to 71.5% in different regions in the past decade".

Response 1: We are sorry for the confusing presentation. We have rewritten the sentence in the revised manuscript: According to other studies performed in China, the proportions of pneumococcal infections in cases of community-acquired pneumonia differ in different regions, varying from 28.0% to 71.5%. (Page 1, Line 41-43)

Point 2: In the second paragraph of the Introduction, I wrote, "The China Antimicrobial Resistant Surveillance System revealed the highly antimicrobial resistance rates regarding Streptococcus pneumoniae in elderly people aged ≥ 65 years". It is important to detail the respective antibiotics and resistance rates.

Response 2: The respective resistance rates to cefuroxime, penicillin, and erythromycin are 46.4%, 53.6%, and as high as 94.2% in older adults aged over 65 years. We have added the information in the revised manuscript. (Page 2, Line 50-51).

Point 3: In the second paragraph of the Introduction, I wrote, "PPSV23 remains exceptionally low in most areas of mainland China when compared to Hong Kong and other developed countries, such as the United States and England with the pneumococcal vaccination coverage as 61.3% and 69.8%, respectively for older adults". Age deciles should be reported as the coverage rate varies by age.

Response 3: Thanks for the comments. We have added the information in the revised manuscript. (Page 2, Line 62-66).

Point 4: In the Study Sites and Participants, I wrote, "The data were collected through an approved questionnaire from...". Datum is singular, while data are plural.

Response 4: The word "was" has been corrected in "were" in the revised manuscript text. (Page 2, Line 88).

Point 5: In the Study Sites and Participants, how many students did you contact, and how many results came back per student: how many grandparents, uncles, aunts…? How many refusals, and how many responses had missing data?

Response 5: Regarding the targeted population, there are a total of 125 598 people aged over 60 years on this study site. We aimed to collect as many samples as possible during the current situation, so we sent the electric questionnaires to 87 primary and secondary schools covering the whole study site, including 187 552 students, rather than sample collection using a sampling method. If there are multiple students in the same family, there is unnecessary to fill in the questionnaire repeatedly according to our guideline. The relationship between the students and the older adults was not presented in the questionnaire. Due to the complex population structure in a family in China, we did not calculate how many older adults in these families; therefore, the number of refusal samples was unknown. The respondents were under quality control, and there was no missing data from the responses. We conducted this survey on the largest online survey platform in China: Wen Juan Xing (Changsha Ranxing Information Technology Co., Ltd., Hunan, China), and only the completed questionnaires would be submitted successfully. We have added the information in the revised manuscript. (Page 2-3, Line 89-114).

Point 6: In the Questionnaire, I wrote, "We developed a self-administered questionnaire which has achieved agreements from epidemiological experts for the current study". Which questions were there disagreement?

Response 6: The current questionnaire comprised four parts regarding general characteristics, health information, knowledge and perception, and vaccination willingness. For the first version of the questionnaire, some general information was re-separated into different levels, including age, education level, and living areas. Regarding the knowledge and perception of pneumonia and its vaccine, the question "Have you ever been received influenza vaccines?" was removed due to experts' disagreement.  

Point 7: What advice do you propose to the government authorities to increase the effectiveness of vaccination campaign programs and uptake?

Response 7: According to our investigation, we consider that the government should cooperate with the communities and the units where the elderly live to strengthen the publicity of the pneumococcal vaccine and its benefits for health. In China, many older persons live together with their children. Hence, the knowledge about pneumococcal pneumonia and its preventive measures should be distributed to every family in a booklet because the family members might encourage older people to accept pneumococcal vaccines. We have added the information in the revised manuscript. (Page 9, Line 304-309).

Reviewer 3 Report

I was invited to revise the paper entitled "Heightened willingness toward pneumococcal vaccination in the elderly population in Shenzhen, China: a cross-sectional study during the COVID-19 pandemic". It aimed to investigate the attitude towards PNC vaccination among elderly Chinese subjects. I want to congratulate with Authors for this paper: it is well presented, easy to read and performed in a right way. In addition, the research topic is of interest for the readers and improve the knowledge in this field. 

Introduction is well written and the background is clearly presented. Methods are adequate and result are well presented and clear to understand for the reader. I have only minor observations:

  • Please report the results of survey validation for each section at least in supplementary materials;
  • sample size estimation is missing;
  • If possible, report also the presence of cancer as comorbidity;
  • in discussion section, compare the results with previous published paper, also from other countries.

Author Response

Thank you for your comments and suggestions that are important to improve our manuscript and play an essential role in our future effort. Please see our point-by-point response below.

Point 1: Please report the results of survey validation for each section at least in supplementary materials.

Response 1: In the present study, the validity of the questionnaire was assessed by experts in epidemiology and infectious diseases. We have supplemented the survey validation in the revised manuscript. (Page 3, Line 118-119).

Point 2: The sample size estimation is missing.

Response 2: Thanks for the comment. In our study, the electric questionnaires were sent to 87 primary and secondary schools covering the whole study site, rather than sample collection using a sampling method. We have added the statement in the revised manuscript. (Page 2, Line 93-95).

Point 3: If possible, report also the presence of cancer as comorbidity.

Response 3: Unfortunately, the cancer information was not collected in this survey. Instead, we included other chronic diseases as a comorbidity in the manuscript text, such as hypertension and diabetes.

Point 4: In the discussion section, compare the results with the previously published paper, also from other countries.

Response 4: We have included this necessary information in the Discussion of the revised manuscript as requested. (Page 7, Line 234-245).